# Cytotoxicity and Antimicrobial Resistance of *Salmonella enterica* Subspecies Isolated from Raised Reptiles in Beijing, China

**DOI:** 10.3390/ani13020315

**Published:** 2023-01-16

**Authors:** Dingka Song, Xuebai He, Yiming Chi, Zhao Zhang, Jing Shuai, Hui Wang, Qiuming Li, Mengze Du

**Affiliations:** 1Animal Science and Technology College, Beijing University of Agriculture, National Demonstration Center for Experimental Animal Education, Beijing 102206, China; 2State Key Laboratory of Oncogenes and Related Genes, Center for Single-Cell Omics, School of Public Health, Shanghai Jiao Tong University School of Medicine, Shanghai 200025, China

**Keywords:** reptiles, *Salmonella* spp., antibiotic resistance, zoonotic disease, cytotoxicity

## Abstract

**Simple Summary:**

Reptiles are well recognized as the asymptomatic carriers of *Salmonella* spp., which is mainly inhabited in the gastrointestinal (GI) mucosa of reptile species. A variety of *Salmonella* serovars, including human-specific pathogenic strains, have been isolated from reptiles previously. In addition, with the growth of the pet reptile market in China, reptile-associated *Salmonella* infections have been noticed as a significant contributor to overall human salmonellosis. However, it remains unclear regarding the prevalence of reptile-associated *Salmonella* in China or its implications on human health. This study aims to investigate the prevalence of *Salmonella* in captive reptile species in the Beijing area through culturation-based identification and to characterize drug resistance as well as host cell virulence in these isolated species. Further, by assessing the overall prevalence of drug-resistant *Salmonella* strains in captive reptiles in Beijing, China, our results highlight the potential threat of zoonotic salmonellosis from pet reptiles in the Beijing area of China.

**Abstract:**

Background: Reptiles are asymptomatic carriers of *Salmonella* spp. Reptile-associated *Salmonella* infections have been noticed as a significant contributor to overall human salmonellosis. However, it remains unclear regarding the prevalence of reptile-associated *Salmonella* in China. Methods: Fecal and gastrointestinal mucosal samples were taken from 104 snakes, 21 lizards, and 52 chelonians and cultured on selective medium. The positive clones were validated and annotated by biochemical screening and multiplex PCR verification. In addition, the antibiotic resistance of identified *Salmonella* isolates was detected and followed by cytotoxic activity detection on human colon cells via co-culturation. Results: The overall prevalence of *Salmonella* in reptiles was 25.99%, with rates of 30.77%, 47.62%, and 7.69% in snakes, lizards, and chelonians, respectively. Further, all isolates showed variable drug-resistant activity to 18 antibiotics, of which 14 strains (30.43%) were resistant to more than eight kinds of antibiotics. More than half of isolated *Salmonella* strains were more toxic to host cells than the standard strain, SL1344. Whole genome sequencing (WGS) results showed that all lizard-associated strains belong to 4 serovar types, and 7 of them fall into the highly pathogenic serovars “Carmel” and “Pomona.” Conclusions: Our results highlight the potential threat of zoonotic salmonellosis from captive reptiles in the Beijing area of China.

## 1. Introduction

The *Salmonella* species are gram-negative bacteria of the *Enterobacteriaceae* family that are major contributors to the global burden of human gastroenteritis [1]. *Salmonella enterica* and *Salmonella bongori* are two species under the genus *Salmonella* [2]. In addition, *Salmonella enterica* is further divided into six subspecies, each with multiple serovars. Of which, the vast majority of serovars are classified under *Salmonella enterica* (2637/2659) [3,4]. The members of *Salmonella enterica subsp. enterica* (I) are primarily responsible for causing diseases in birds, mammals, and even humans. In other *S. enterica subspecies,* it has been linked to isolated or sporadic diseases in both mammals and reptiles [2]. According to statistics from the World Health Organization (WHO), *Salmonella* is the major cause of gastroenteritis in humans [5], and 1.35 million cases of *Salmonella* infection were reported in the United States annually [6]. The most common serovar types for human infection are *S. enterica serovar Enteritidis* and *S. enterica serovar Typhimurium*, which induce GI symptoms or fever, respectively [7]. Direct contact with live animals or animal products is the major route of *Salmonella* infection [8]. The clinical symptoms of *Salmonella* infection include diarrhea, vomiting, and fever, which may cause life threatening septicemia in serious cases, while children and elder populations are highly vulnerable to *Salmonella* infection [8,9]. 

Reptiles are the natural reservoir of variable serovar types of *Salmonella* (some of which have been known to be pathogenic to humans) with few or no symptoms [10,11,12]. Notably, snakes, lizards, and turtles have been identified as the natural hosts of *Salmonella* through traceback investigation of both individual cases and outbreaks worldwide [10,13,14,15,16,17]. In China, it was also reported that highly pathogenic *Salmonella* strains exist in the wild in the red-eared slider (*Trachemys scripta elegans*) [18]. In recent years, the trend of pet reptile ownership has become increasingly popular worldwide [19], and antibiotics have been widely used in the breeding, housing, and transportation of pet reptiles for the purposes of animal welfare and economic preservation [20,21]. However, overuse of antibiotics led to the outgrowth of multi-drug resistant (MDR) *Salmonella* strains and brought major concerns for public health safety [8,22]. 

Currently, tortoises and freshwater turtles can be raised by individuals for pet use in China with authorization. However, the concerns regarding the emergence of MDR *Salmonella* strains in pet reptiles and the risk of reptile-associated salmonellosis in humans have not been fully revealed.

## 2. Materials and Methods

### 2.1. Ethics Statement

The animal study was reviewed and approved by the Animal Ethics Committee of the Beijing University of Agriculture under the protocol BUA2022071. All the pathogens used in this study strictly complied with *the Regulations on Biological Safety Management of Pathogen Microbiology Laboratory* (000014349/2004-00195) from the State Council of the People’s Republic of China.

### 2.2. Animal Selection and Sampling

Fresh fecal and gastrointestinal mucosal samples were taken from five reptile breeders or commercial farms in Beijing for *Salmonella* examination during 2021.10–2022.1. The exclusion criteria for animals are: (1) Showing signs of disease. (2) treated with antibiotics within the last 45 days; and (3) shared a cage with reptiles treated with antibiotics in the last 30 days. (4). reptiles shedding or taking food (when animals were hypersensitive to external stimulations) [23]. All samples collected from lizards were fecal, while all cloacal swab samples were collected for snakes and chelonians. A total of 23 reptile species covering snakes, lizards, and chelonians were screened, and 177 samples were collected, including 104 from snakes, 21 from lizards, and 52 from chelonians. The annotation for the type and number of reptiles sampled is described in Table 1. All animals were considered healthy at the time of sampling based on daily observations by the breeders during the previous month and a physical examination by a veterinarian at the time of sampling.

### 2.3. Sample Collection and Processing

A sterilized, soft swab of appropriate size was used for collecting gastrointestinal mucosal samples or fresh feces (within 6 h). In addition, for cloacal swab collection, animals were physically restrained, and an applicable swab was inserted into the cloaca and gently rotated longitudinally. Further, for onsite isolation, swabs were immediately plated on modified semi-solid rappaport-vassiliadis (MSRV) medium (Beijing Land Bridge, Beijing, China) and cultured at 42 ± 1 °C for 18~24 h [24]. The suspected colonies were transferred onto xylose lysine tergitol 4 (XLT4) agar (Beijing Land Bridge, Beijing, China) and cultured at 37 ± 1 °C for another 18~24 h. The colonies with the “middle black” feature were regarded as possible *Salmonella* isolates and were sub-cultured in XLT4 agar for an additional 2~3 generations. In the lab, swabs with cloacal or fecal materials were diluted in 10 mL buffered peptone water (Beijing Land Bridge, Beijing, China) to pre-enrichment at 37 ± 1 °C for 18 ± 2 h. The isolation was performed onsite immediately after the collection of each sample in this study. All samples were streaked with suspected *Salmonella* colonies onto urea agar (Beijing land bridge, Beijing, China) and triple sugar iron (TSI) agar (Beijing land bridge, Beijing, China) and incubated at 37 ± 1 °C for 24 h. Isolates with a negative urea reaction and the production of hydrogen sulfide in the TSI test were primarily considered *Salmonella*. The screened colonies were sequentially verified by 16s rDNA sequencing [25]. In the sequence analysis, 16s rDNA sequences were aligned with standard databases via BLASTn at NCBI (https://blast.ncbi.nlm.nih.gov/Blast.cgi (accessed on 12 December 2021 and 20 March 2022)). The matches with certain criteria: (percent identity > 95% and e_value less than 1 × 10^−50^) were taken into consideration. In each sequence, results from the BLASTn algorithm were parsed to keep only the first best match. A total of 16s sequencing-verified *Salmonella* strains were propagated in trypticase soy broth medium (Beijing Land Bridge, Beijing, China) and stored at −80 °C with 25% glycerol for later use.

### 2.4. Multiplex PCR Assay 

The bacterial genomic DNA was then extracted after overnight culturation in trypticase soy broth medium (Beijing Land Bridge, Beijing, China) by using a QIAGEN genomic DNA purification kit (Tiangen Biotech, Beijing, China) and stored at −80 °C.

The target genes were chosen according to previous studies [26], which include *flijB*, *mdcA*, *gatD*, *stn*, STM4057, and *invA.* Each multiplex PCR tube contained 0.3 mmol/L of each deoxyribonucleotide triphosphate, 1 × Ex Taq Buffer (Takara Biomedical Technology, Beijing, China), a *stn* primer pair (0.75 μmol/L), a *fljB*, *mdcA*, *gatD,* and *invA* primer pair (0.50 μmol/L), or 1 each STM4057 primer pair (0.25 μmol/L), 2.0 μL template DNA, and 0.4 U of TaKaRa Ex TaqTM Hot Start Version (Takara Biomedical Technology, Beijing, China). The volume was adjusted with sterile distilled water to 20 μL. A PCR reaction was carried out in a PCR amplifier under the following conditions: denaturation at 95 °C for 10 min, followed by 40 cycles of amplification (denaturation at 95 °C for 1 min, annealing at 60 °C for 1 min and extension at 72 °C for 1 min), ending with a final extension at 72 °C for 15 min. The amplified products were separated by electrophoresis on 2.5% agarose gels in 1 × Tris-acetate-EDTA buffer (Solarbio, Beijing, China) using a DYCP-31C electrophoresis system (Liuyi Biotechnology, Beijing, China), stained with ethidium bromide, visualized under UV irradiation, and photographed with a 3UV transilluminator NLMS-20E (Atto, Tokyo, Japan).

### 2.5. Evaluation of Cytotoxic Activity on Human Cells

A cytotoxicity assay was performed based on the Enhanced Cell Counting Kit-8 (CCK-8) (Beyotime, Beijing, China). In order to detect the cell mortality induced by reptile *Salmonella*, Caco-2 cells (5000 cells/well) were seeded into a 96-well cell culture plate and treated with reptile *Salmonella* (1 × 10^4^ CFU, MOI = 2; or 2.5 × 10^4^ CFU, MOI = 5). 24 h after co-culturation, cells were washed with PBS and cultured in DMEM supplemented with 2% FBS and 10 µL of enhanced CCK-8 solution. The OD450 was measured after 2 h of incubation. Further, the MOI (multiplicity of infection) indicates the ratio of bacterial number to cell number. 

### 2.6. Antimicrobial Resistance Evaluation

The minimum inhibitory concentrations (MICs) of ampicillin, meropenem, amoxicillin-clavulanic acid, ceftiofur, cefazolin, gentamicin, streptomycin, amikacin sulfate, kanamycin, ciprofloxacin, enrofloxacin, nalidixic acid, chloramphenicol, florfenicol, tetracycline, polymyxin B sulfate, macrodantin, and bactrim were determined according to performance standards for antimicrobial susceptibility tests compiled by the Clinical and Laboratory Standards Institute [27]. Specifically, *Salmonella* strains were inoculated in CAMH broth, cultured at 37 ± 1 °C and 200 rpm for 12 h, then diluted to 5 × 10^5^ CFU/mL in CAMH broth. The concentrated antibiotic stock solutions were serially diluted in a sterile 96-well cell culture plate (100 μL per well). A diluted *Salmonella* suspension was then added into each well with diluted antibiotic solutions (100 μL per well). The CAMH broth was served as a negative control, and *Escherichia coli* ATCC 25922 was used for quality control. In addition, bacteria and drugs were thoroughly mixed and incubated at 37 ± 1 °C for 16~18 h before data acquisition. Three parallel tests were performed for each antibiotic. The results were judged in accordance with the standards of the Clinical and Laboratory Standards Institute (CLSI). However, the lowest concentration of drug that sufficiently inhibits bacterial growth was taken as the minimum inhibitory concentration (MIC) (unit: μg/mL). The isolates were classified as susceptible or resistant according to their MICs for a given drug. See Appendix A for the specific judgment criteria and the MICs of each strain for the antibiotics designed in this experiment.

### 2.7. Whole Genome Sequencing (WGS)

The bacterial isolates were recovered and cultured from stocks. The DNA was extracted using a bacteria DNA isolation kit (Tiangen Biotech, Beijing, China). The draft genome sequencing was performed at Personal Gene Technology Co., Ltd. (Nanjing, China). In addition, the Illumina NovaSeq instrument (Illumina, San Diego, CA, USA) was used for the genome sequencing with a 150-bp paired-end strategy. The A5-miseq and SPAdes were used to perform quality trimming and de novo assembly of the reads [28,29]. Further, all raw reads have been uploaded to the NCBI database (Bioproject Accession No. PRJNA922043). The serotype, and antimicrobial resistance gene detection were performed using the Center for Genomic Epidemiology server (https://cge.cbs.dtu.dk (accessed on 1 July 2022)).

### 2.8. Statistical Analysis

A Kruskal-Wallis ANOVA with post hoc analysis utilizing Dunn’s multiple comparisons test was used to determine the statistical significance of the cytotoxic activity of *Salmonella* strains in Caco-2 cells. All statistical analyses were performed using GraphPad Prism (version 7, GraphPad Software Inc., San Diego, CA, USA). *P*-values less than 0.05 were considered statistically significant.

## 3. Results

### 3.1. Prevalence of Salmonella spp. in Screened Reptiles

A total of 46 out of 177 samples were *Salmonella*-positive, as determined by biochemical tests (Table 1). In snakes and lizards, 30.77% and 47.62% of samples were positive for *Salmonella*, respectively. In Chelonians, *Salmonella* was only identified in 4 samples (7.69%). The prevalence of *Salmonella* was markedly lower in chelonians than in lizards and snakes, which is consistent with previous studies [23].

The multiplex PCR analysis revealed that all isolated *Salmonella* strains belong to the species *Salmonella. enterica*, and are composed of 4 subspecies: I, IIIb, IV, and V, while subspecies I and IIIb are the dominant (43.47% and 34.78%, respectively) (Table 2).

### 3.2. Cytotoxicity of Isolated Salmonella spp.

In order to investigate the potential impacts of *Salmonella* on host cells, an in vitro cytotoxic assay was performed by co-culturation of *Salmonella* isolated from reptiles with Caco-2, a human colon cell line. The *Salmonella Typhimurium* 1344 (SL1344) was included as a control. Further, cell viability following co-culturation was detected by CCK-8 staining. It showed that the cytotoxicity of *Salmonella* isolates on Caco-2 cells was highly variable (Figure 1). Notably, 60.86% of isolates showed stronger cytotoxicity than the standard strain SL1344 at a multiplicity of infection (MOI) = 2, mainly composed of isolates in subspecies I and IIIb. The result was similar, with an MOI = 5. The high-virulence strains were defined as those showing higher cytotoxicity than SL1344 in both MOIs. Specifically, subspecies I and IIIb are predominantly found in high virulence strains. Due to the low isolating rate, it is not possible to evaluate the prevalence of high-virulence strains in other species than I and IIIb. In addition, 50% of strains isolated from turtles exhibited cytotoxicity higher than that of SL1344 (2/4), the percentages were 70% (7/10) and 34.37% (11/32) for strains isolated from lizards and snakes in both MOIs, respectively.

### 3.3. Serovar Information of Salmonella Isolated from Lizards via WGS

Given that the most virulent *Salmonella* strains in subspecies I predominantly existed in lizards (*Pogona vitticeps*), we sought to further investigate the serovar information of these strains via WGS. A total of 4 serovar types were identified, of which the serovar type “carmel” is most dominant (up to 50% of all tested strains); additionally, “ago,” “pomona,” and “IIIb 57: c:z” were also identified, indicating the complex composition of serovar types in lizards (Table 3). 

### 3.4. Antibiotic Resistant Profiles of Reptile Associated Salmonella Isolates 

In order to investigate the vulnerability of newly isolated *Salmonella* strains from reptiles in response to antibiotics, we also performed a drug-resistant assay by treating *Salmonella* isolates with various kinds and doses of antibiotics. It showed that 46 *Salmonella* isolates (100%) were resistant to at least 3 types of antibiotics, and 25 isolates were resistant to at least 6 antibiotics (Appendix A). These results were comparable or even higher than a previous study in Australia, which showed that among 92 strains of *Salmonella,* only two exhibited prominent antibiotic resistance [30]. Specifically, a major proportion of *Salmonella* isolates were resistant to macrodantin (97.87%), tetracycline (91.47%), gentamicin (89.36%), kanamycin (51.06%), and ciprofloxacin (51.06%), and the least resistant antibiotics were amikacin sulfate (2.13%) and meropenem (0%) (Table 4). The isolates from lizards, snakes, and chelonians were 53.12% (17/32), 50% (5/10), and 75% (3/4) of *Salmonella* isolates were resistant to at least 6 types of antibiotics, respectively (Appendix A). 

## 4. Discussion

It is well known that reptiles carry a large amount of *Salmonella,* which can be spread into the environment. It has been surveyed for the prevalence of *Salmonella* in different reptile species covering various countries or regions worldwide, including Japan, Germany, Austria, Italy, Australia, Norway, New Zealand, Croatia, etc. [23,31,32,33,34,35]. Meanwhile, only a few studies investigating the prevalence of *Salmonella* in reptiles in China have been conducted. It is notable that China has plentiful reptile resources, and raising reptiles for pet use is becoming a trend in recent years [36]. However, the concern of reptile associated zoonotic salmonellosis has not been well addressed. Therefore, it is worthwhile to understand the distribution and characteristics of *Salmonella* species in raised reptiles. In this study, it was shown that lizards and snakes were more likely to carry *Salmonella* than turtles in Beijing, which is consistent with previous studies in other countries [31,32,37]. However, other studies hypothesiz that turtles may have a higher rate of *Salmonella* infection [35,38,39]. The geographic difference may explain the controversial observations. Nevertheless, due to the limited sample size and restricted types of turtles (mainly aquatic) involved in this study, it may compromise the accuracy of *Salmonella* prevalence in turtles. For instance, *InvA*, located on *Salmonella* Pathogenicity Island 1 (SPI-1), is prevalently found in *Salmonella* species and is also well-known as an invasion gene [40]. It has been established by the U.S. Food and Drug Administration as a confirmatory gene for pathogenic *Salmonella* spp. [41] and was also taken by us as one of the marker genes for *Salmonella* identification in the multiplex PCR assay. However, parts of the stains showed a negative PCR result for the *invA* gene in our study. According to previous studies, Kadry et al. revealed that in eight *Salmonella* isolates, only 50% were positive for the *invA* gene in both egg and human isolates [42]. Similarly, a study of *Salmonella* from milkfish in Indonesia showed that *invA* was found in only 12.5% of all sampled aquatic products [43]. Therefore, we speculated that the *Salmonella* strains isolated from our reptile samples were similar to those in the aquatic products. Nevertheless, it needs to be validated by the following studies. Further, another limitation of our study is that the results were obtained by analyzing samples from certain places in the Beijing area, which may not accurately reflect the overall prevalence of *Salmonella* in Beijing. Nevertheless, our work provides valuable insights for systemic retrospective studies in the future. 

The cytotoxic assay is a conventionally used strategy to access the pathogenicity of bacteria in vitro, which is obtained by comparing the percentage of live cells with/without bacteria co-incubation. In our study, SL1344 was used as an external control to evaluate the virulence of *Salmonella* isolates. To our surprise, approximately half of the isolated strains showed higher cytotoxicity than SL1344, which has been demonstrated to be able to cause diarrhea in cattle [44]. These results were consistent with previous reports that *Salmonella enterica* subspecies I is highly pathogenic [32]. Among all tested isolates, the most pathogenic strain was 1101PV5, which belongs to the serovar type of Pomona, which was consistent with previous pre-clinical and clinical studies on reptile associated Salmonellosis [18,45,46].

According to previous studies, the prevalence of drug resistance in reptile-associated *Salmonella* was relatively low. In vitro antibiotic resistance assays revealed that the percentage of MDR strains observed in reptiles was 0–14% [47,48]. On the contrary, a large number of MDR strains were isolated in our study. Therefore, rather than a validation of previous studies on reptile associated *Salmonella* in different countries/regions, our study is closer to the results of *Salmonella* antibiotic profiling on poultry products or live poultry markets in China. It is reported that the resistance rates of *Salmonella* strains against ampicillin, tetracycline, and colistin in live birds were as high as 97.6%, 58.3%, and 51.2%, respectively [49]. Additionally, a nationwide survey of *Salmonella* from eggs showed that the antibiotic resistance rate of *Salmonella* strains was as high as 64.3%, 39.3%, and 21.4% against nalidixic acid, ampicillin, and tetracycline, respectively [50]. Therefore, it is implicated that exposure to potentially toxic antibiotics (in the same area) plays a more critical role in the development of antibiotic-resistant *Salmonella* features than the nature of hosts. These results were obtained by in vitro experiments, while the antibiotic-resistant capacity of *Salmonella* during intracellular infection increases dramatically [51]. Therefore, in vivo studies are still required to further investigate the effect of antibiotics on *Salmonella* during infection. 

In our study, antibiotics for which the resistance rate of *Salmonella* isolates is over 40% include macrodantin (97.87%), tetracycline (91.49%), gentamicin (89.36%), kanamycin (51.06%), ciprofloxacin (51.06%), and cefazolin (44.68%). Surprisingly, it was found that the resistance rate to polymyxin was as high as 27.66%. According to on-site veterinarians, only a limited number of types of antibiotics were used by the breeders (including Enrofloxacin, Gentamicin, and Cephalosporin; no antibiotics were used on lizards). However, the prevalence of drug resistance for isolated *Salmonella* strains is unprecedentedly high, while the clues for the development of antibiotic resistance remain unclear. One possible explanation is that *Salmonella* can persist in the GI lumen of healthy reptiles, which facilitates horizontal gene transfer (HGT) of antibiotic-resistance genes to *Salmonella* from other symbiotic bacterial species or from the environment, which needs to be addressed by additional studies. Furthermore, given the fact that few known antibiotic-resistant genes have been identified by WGS analysis (data not shown), it is possible that unknown genes are responsible for the development of an antibiotic resistant phenotype in newly identified *Salmonella* strains. Further, to pinpoint and validate the critical genes, additional approaches, including forward mutagenesis screening, targeted mutation, and functional assays, will need to be performed in the future [52]. 

## 5. Conclusions

The study involved the collection and characterization of *Salmonella spp.* from multiple reptile breeders in Beijing, China. In addition, the study demonstrates that human-raised reptiles are carriers of potential zoonotic *Salmonella*. However, the persistence of MDR *Salmonella* strains in reptiles would undoubtedly become a “pool” that preserves an accumulating number of drug-resistant genes, thereby accelerating the spread of drug-resistant genes in the environment and eventually causing a serious public health crisis. It is important to point out that reptiles in general carry *Salmonella*; treatment in healthy reptiles is not always necessary but might lead to the development of drug-resistant *Salmonella* colonies, which is a huge threat to human health. On the basis of these facts, it is emphasized that appropriate policies for breeding and antibiotic treatment of reptiles should be carried out in time to prevent the development of MDR *Salmonella* in reptiles or potential zoonotic salmonellosis in humans. 

## Figures and Tables

**Figure 1 animals-13-00315-f001:**
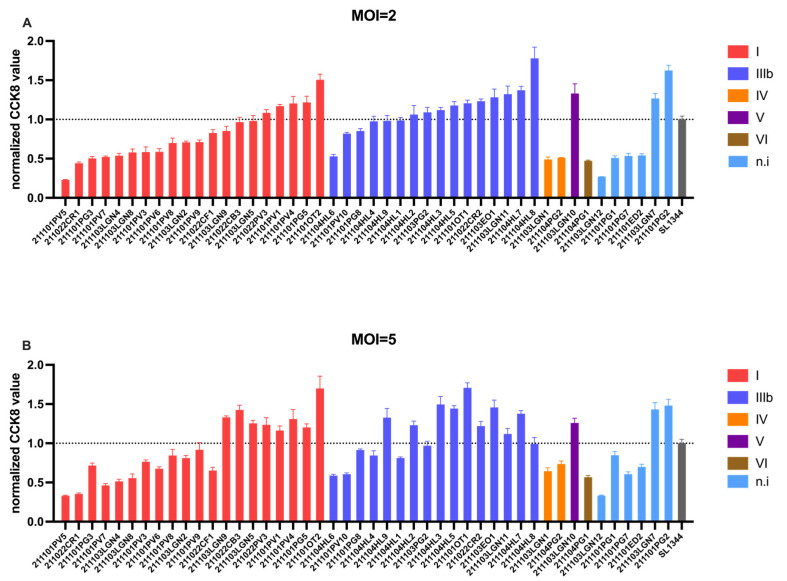
Cytotoxic capacity of *Salmonella* strains isolated from reptiles. 46 *Salmonella* strains isolated from reptiles were tested for their cytotoxicity on Caco-2 cells at MOI = 2 (**A**) and MOI = 5 (**B**). *Salmonella Typhimurium* SL1344 was included as a control.

**Table 1 animals-13-00315-t001:** Prevalence of *Salmonella spp.* in reptiles via culture-based screening.

Type	Species	Common Name	No. Positive/No. Tested (%)
Turtle	*Cuora amboinensis*	Amboina Box Turtle	0/1
	*Cuora aurocapitata*	Yellow-headed Box Turtle	0/1
	*Cuora flavomarginata*	Snake-eating Turtle	1/19
	*Cuora galbinifrons*	Indochinese Box Turtle	0/1
	*Cuora mccordi*	McCord’s Box Turtle	0/2
	*Cuora mouhotii*	Keeled Box Turtle	0/1
	*Cuora pani*	Pan’s Box Turtle	0/1
	*Cuora trifasciata*	Chinese Three-striped Box Turtle	0/2
	*Cuora yunnanensis*	Yunnan Box Turtle	0/1
	*Mauremys japonica*	Japanese Pond Turtle	0/1
	*Mauremys reevesii*	Reeves’ Turtle	3/20
	*Mauremys sinensis*	Chinese Striped-necked Turtle	0/1
	*Trachemys scripta elegans*	Red-eared Slider	0/1
	Total		4/52 (7.69)
Snake	*Elaphe dione*	Steppes Ratsnakes	1/5
	*Elaphe taeniura*	Beauty Snake	2/5
	*Heterodon nasicus*	Hognose Snake	9/15
	*Gonyosoma frenatum*	Khasi Hills Trinket snake	0/1
	*Pantherophis guttatus*	Corn Snake	9/33
	*Pantherophis obsoletus*	Rat Snake	1/3
	*Lampropeltis getula*	Kingsnake	10/41
	*Lycodon rufozonatus*	Red-banded Snake	0/1
	Total		32/104 (30.77)
Lizard	*Pogona vitticeps*	Bearded Dragon	10/19
	*Rhacodactylus leachianus*	New Caledonia Giant Gecko	0/2
	Total		10/21 (47.62)
Total			46/177 (25.99)

Table 1 shows the number of different types of reptiles included for sampling as well as the number of positive *Salmonella* strains isolated. Summaries of numbers and percentages for different types of reptiles are listed at the end of each sub-row.

**Table 2 animals-13-00315-t002:** Subspecies information of *Salmonella* isolates based on Multiplex PCR.

			Multiplex PCR Result						Subspecies
Strain ID	Collecton Sites	Reptile Source	*fljb*	*mdcA*	*gatD*	*stn*	STM4057	*invA*	
1101ED2	A (Individual breeder)	*Elaphe dione*						+	n.i
1101OT1	A (Individual breeder)	*Elaphe taeniura*	+	+		+			IIIb
1101OT2	A (Individual breeder)	*Elaphe taeniura*			+	+		+	I
1101PG1	A (Individual breeder)	*Pantherophis guttatus*		+					n.i
1101PG2	A (Individual breeder)	*Pantherophis guttatus*		+					n.i
1101PG3	A (Individual breeder)	*Pantherophis guttatus*			+	+	+	+	I
1101PG5	A (Individual breeder)	*Pantherophis guttatus*			+	+	+		I
1101PG7	A (Individual breeder)	*Pantherophis guttatus*		+					n.i
1101PG8	A (Individual breeder)	*Pantherophis guttatus*	+	+		+			IIIb
1103EO1	B (Individual breeder)	*Pantherophis obsoletus*	+	+		+			IIIb
1103LGN1	B (Individual breeder)	*Lampropeltis getula*				+			IV
1103LGN10	B (Individual breeder)	*Lampropeltis getula*			+	+			V
1103LGN11	B (Individual breeder)	*Lampropeltis getula*	+	+		+			IIIb
1103LGN12	B (Individual breeder)	*Lampropeltis getula*		+					n.i
1103LGN2	B (Individual breeder)	*Lampropeltis getula*	+		+	+	+	+	I
1103LGN4	B (Individual breeder)	*Lampropeltis getula*			+	+	+		I
1103LGN5	B (Individual breeder)	*Lampropeltis getula*			+	+	+		I
1103LGN7	B (Individual breeder)	*Lampropeltis getula*			+	+			n.i
1103LGN8	B (Individual breeder)	*Lampropeltis getula*			+	+	+		I
1103LGN9	B (Individual breeder)	*Lampropeltis getula*	+		+	+	+	+	I
1103PG2	B (Individual breeder)	*Pantherophis guttatus*	+	+		+		+	IIIb
1104HL1	C (Individual breeder)	*Heterodon nasicus*	+	+		+			IIIb
1104HL2	C (Individual breeder)	*Heterodon nasicus*	+	+		+		+	IIIb
1104HL3	C (Individual breeder)	*Heterodon nasicus*	+	+		+		+	IIIb
1104HL4	C (Individual breeder)	*Heterodon nasicus*	+	+		+			IIIb
1104HL5	C (Individual breeder)	*Heterodon nasicus*	+	+		+		+	IIIb
1104HL6	C (Individual breeder)	*Heterodon nasicus*	+	+		+			IIIb
1104HL7	C (Individual breeder)	*Heterodon nasicus*	+	+		+		+	IIIb
1104HL8	C (Individual breeder)	*Heterodon nasicus*	+	+		+		+	IIIb
1104HL9	C (Individual breeder)	*Heterodon nasicus*	+	+		+			IIIb
1104PG1	C (Individual breeder)	*Pantherophis guttatus*	+		+	+		+	VI
1104PG2	C (Individual breeder)	*Pantherophis guttatus*				+		+	IV
1022PV3	D (Individual breeder)	*Pogona vitticeps*			+	+	+		I
1101PV1	D (Individual breeder)	*Pogona vitticeps*	+		+	+	+	+	I
1101PV10	D (Individual breeder)	*Pogona vitticeps*	+	+		+		+	IIIb
1101PV3	D (Individual breeder)	*Pogona vitticeps*			+	+	+	+	I
1101PV4	D (Individual breeder)	*Pogona vitticeps*	+		+	+	+	+	I
1101PV5	D (Individual breeder)	*Pogona vitticeps*	+		+	+	+	+	I
1101PV6	D (Individual breeder)	*Pogona vitticeps*	+		+	+	+	+	I
1101PV7	D (Individual breeder)	*Pogona vitticeps*	+		+	+	+	+	I
1101PV8	D (Individual breeder)	*Pogona vitticeps*	+		+	+	+	+	I
1101PV9	D (Individual breeder)	*Pogona vitticeps*	+		+	+	+	+	I
1022CB3	E (Commercial farm)	*Mauremys reevesii*	+		+	+	+	+	I
1022CF1	E (Commercial farm)	*Cuora flavomarginata*	+		+	+	+	+	I
1022CR1	E (Commercial farm)	*Mauremys reevesii*	+		+	+	+	+	I
1022CR2	E (Commercial farm)	*Mauremys reevesii*	+	+		+			IIIb

Table 2 lists all *Salmonella* strains isolated from reptiles and the PCR results for six marker genes. Subspecies information about isolates was obtained based on the positive/negative results of each gene. N.i. non-identified.

**Table 3 animals-13-00315-t003:** Serovar information of *Salmonella* isolates based on WGS.

Strain ID	Reptile Source	Subspecies	Serovar Type
1022PV3	*Pogona vitticeps*	I	ago
1101PV1	*Pogona vitticeps*	I	pomona
1101PV10	*Pogona vitticeps*	IIIb	IIIb 57:c:z
1101PV3	*Pogona vitticeps*	I	ago
1101PV4	*Pogona vitticeps*	I	pomona
1101PV5	*Pogona vitticeps*	I	carmel
1101PV6	*Pogona vitticeps*	I	carmel
1101PV7	*Pogona vitticeps*	I	carmel
1101PV8	*Pogona vitticeps*	I	carmel
1101PV9	*Pogona vitticeps*	I	carmel

Whole genome sequencing of *Salmonella* isolates from lizards (*Pogona vitticeps*) identified 4 serovar types, including “ago”, “pomona,” “carmel,” and “IIIb 57:c:z”.

**Table 4 animals-13-00315-t004:** Drug resistant rate of *Salmonella* strains isolated from reptiles in this study.

	*Salmonella* from Snakes (%)	*Salmonella* from Lizards (%)	*Salmonella* from Turtles (%)	Total (%)
Beta-lactam antibiotic				
Ampicillin	25.00	30.00	0	25.53
Meropenem	0	0	0	2.13
Amoxicillin-clavulanic acid	25.00	50.00	0	29.79
Cephalosporin Antibiotic				
Ceftiofur	12.50	10.00	0	14.89
Cefazolin	50.00	40.00	0	44.68
Aminoglycoside antibiotic				
Gentamicin	96.88	70.00	75.00	89.36
Streptomycin	9.38	10.00	25.00	12.77
Amikacin sulfate	0	0	0	2.13
Kanamycin	40.63	60.00	75.00	51.06
Quinolone antibiotics				
Ciprofloxacin	50	40.00	50.00	51.06
Enrofloxacin	18.75	40.00	25.00	25.53
Nalidixic acid	28.13	30.00	0	27.66
Chloramphenicol antibiotics				
Chloramphenicol	3.13	10.00	0	6.38
Florfenicol	15.63	40.00	0	23.40
Tetracycline antibiotics				
Tetracycline	90.63	100.00	75.00	91.49
Polypeptide antibiotics				
Polymyxin B sulfate	25.00	20.00	50.00	27.66
Nitrofuran antibiotics				
Macrodantin	100	90.00	75.00	97.87
Sulfonamide antibiotics				
Bactrim	15.63	40.00	75.00	27.66

Table 4 summarizes the antimicrobial resistance rate of all isolated *Salmonella* strains against a various type of antibiotics. Different classes of antibiotics including beta-lactam, cephalosporin, aminoglycoside, quinolone, chloramphenicol, tetracycline, polypeptide, nitrofuran and sulfonamide were used in this study.

## Data Availability

The data presented in the study are available in the article.

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
