# Peer review of "Cytotoxicity and Antimicrobial Resistance of Salmonella enterica Subspecies Isolated from Raised Reptiles in Beijing, China"

_animals, 2023, doi:10.3390/ani13020315_

Round 1

Reviewer 1 Report

This study contributes to an important an emerging problem. Overall the study design is clear. I have however some remarks and questions to be addressed:

L77-: This is rather a summary and does not belong into the introduction

L95-: In how many cases fecal samples were used? This might be difficult in reptiles, but of relevance for the detection 

L99: what does that mean?

L106-: In other studies, repeated enrichment methods were used and described as necessary to determine the prevalence. You did not use a liquid enrichment boullion before plating. Why?

L139-: this needs to be explained more in detail. What do you mean with MOI (later on), and also details on the assessment and the use of the reference strain (why this one?) are necessary

L147-: there are no MIC values for reptiles. Which values did you use? You should also consider that this is difficult to interpret and the in vitro determination especially in reptiles might differ from the in vivo situation

L147-: describe the method used for sensitivity testing

L174: statistic used is not mentioned

Table 1: it would be good to include the origin of the samples and the collection. Especially since you mention differences later, and the detection might be collection-specific.

Table 2: I am not sure if this detailed information is necessary. It could be summarized for subspecies. The strain IDs are relevant for figure 1, and could be referred to there

Table 3: Here it is necessary to give information on the sampling site/collection. Are all bearded dragons form the same collection?

L 222: various kinds and doses of antibiotics - this should bei explained in M&M

L231: this belongs to the discussion

L250-: I would be careful with the conclusion on the prevalence. You need to include the samplings sites into the statistical evaluation and interpretation. 

L267-: please consider that the data you collected are in vitro resistencies and need to be explained in the context of MIC determination

L299: a conclusion should also be that reptiles in general carry salomenlla and that treatment in healthy reptiles does not make sense but might lead to resistancies. these can then be a problem in humans

Reviewer 2 Report

Comments to the Author

Song et al. investigated antimicrobial resistant Salmonella isolated from raised reptiles in Beijing, China. The antibiotic resistance of identified Salmonella isolates was detected, followed by the detection of cytotoxic activity on human colon cells. The method illustrated in the manuscript needs to be revised and further clarified. WGS data could be further analyzed and discussed.

Below are more specific comments:

Line 95: “…from 5 reptile breeders”: The breeders codes could be specified in method. e.g., “named in sequence from A-E”

Line 109: “…Swabs were immediately plated on MSRV medium…” : Does the MSRV medium indicate the Modified Semi-Solid Rappaport-Vassiliadis (MSRV) medium? Please add the product information. And could you explain why you use MSRV as a selective culturing step? Why is the cultured temperature set at 37℃ rather than 42℃?

Line 109: Could you explain the reason why you skip the pre-enrichment step (usually with PBW) that is usually applied during Salmonella isolation?

Line 117-118: It is not clear what kind of "PCR" and/or "16s rDNA sequencing" the author applied? Is there any reference? if so, could you cite it/them?

Line 120: “…50% glycerol…”: Could it be a typing error? Is the final concentration around 25%?

Line 154: The reads raw data was not available; the author could upload to the European Nucleotide Archive (ENA) or the National Center for Biotechnology Information (NCBI).

Line 182: These data can be enclosed as a supplementary table to be accessible for readers.

Table 2: According to the method description, the subspecies classification is based on the method developed in reference paper 24. However, the multiplex PCR result and the subspecies seem not to be correlated. e.g., according to reference 24, the invA gene is commonly used as a marker to detect Salmonella, which exists in all Salmonella subspecies. However, parts of the stains in Table 2 had a negative PCR result for the invA gene. Could you please explain it?

Figure 1: The X-axis order in Figure 1B should be the same as in Figure 1A, so that readers can easily compare the cytotoxicity of different MOIs. And it is hard for readers to get the percentage of "high virulent strains", Different subspecies could be applied with different color legends.

Line 200-201: The author should define the "high virulent strain" first (according to the contents, it seems like the strains that showed a higher CCK8 value than the value of the SL1344 strain were defined as high virulent strains).

Line 209: Did the author also do the agglutination tests with antisera to confirm the serotype identification?

Line 231-233: Please further explain why this finding could support the deduction.

Line 261: “…showed higher virulence than…”: According to the result, it would be better to use "...showed higher cytotoxicity than..."

Line 288-290: According to the method description, the author sequenced the WGS of some isolates before identifying antibiotic-resistant genes in silico. Why is no result presented in the manuscript? These data could be discussed together with the hypothesis of HGT.

Round 2

Reviewer 1 Report

no further comments, main concerns have been addressed.

Author Response

Many thinks to your comments on our manuscript. This has greatly  helped to improve the quality of our manuscript.

Reviewer 2 Report

Line 117-120:  What is the exact BLAST-positive meaning? How do you calculate the "confidence"? According to my knowledge, it could be BLAST the amplified sequences to the NCBI nr database to see if they match (with high identity, e.g. >95%) to one or some Salmonella 16S ribosomal DNA and at the same time have a low or no match to the 16S ribosomal DNA of other bacterial species. Please clearly describe the analysis steps, as it had been use as the step of “… eventually considered as Salmonella…”.

Line 117-120 and Table 2: Could you please present me some 16s rDNA sequencing data (no reads needed, just the final sequence), especially for the strains that showed 'stn' negatives, e.g. 1101PG1, 1101PG, 7 and 'invA' negatives, e.g. 1101OT1, 1101PG5.

Line 170: Note to both the author and editor: The reads raw data was not available; the author could upload it to the European Nucleotide Archive (ENA) or the National Center for Biotechnology Information (NCBI).
